# Can the Seed Trade Provide a Potential Pathway for the Global Distribution of Foliar Pathogens? An Investigation into the Use of Heat Treatments to Reduce Risk of *Dothistroma septosporum* Transmission via Seed Stock

**DOI:** 10.3390/jof9121190

**Published:** 2023-12-13

**Authors:** Katherine Tubby, Jack Forster, Martin Mullett, Robert Needham, Olivia Smith, James Snowden, Shelagh McCartan

**Affiliations:** 1Forest Research, Forestry Commission, Alice Holt Lodge, Farnham GU10 4LH, UKmartin.mullett@mendelu.cz (M.M.); olivia.smith@forestresearch.gov.uk (O.S.); james.snowden@forestresearch.gov.uk (J.S.); shelagh.mccartan@maelor.co.uk (S.M.); 2Department of Forest Protection and Wildlife Management, Faculty of Forestry and Wood Technology, Phytophthora Research Centre, Mendel University in Brno, Zemědělská 3, 613 00 Brno, Czech Republic

**Keywords:** Dothistroma needle blight, invasive forest pathogens, biosecurity, plant disease, seed quality, phytosanitary

## Abstract

The international plant trade results in the accidental movement of invasive pests and pathogens, and has contributed significantly to recent range expansion of pathogens including *Dothistroma septosporum.* Seeds are usually thought to present a lower biosecurity risk than plants, but the importation of *Pinus contorta* seeds from North America to Britain in the mid-1900s, and similarities between British and Canadian *D. septosporum* populations suggests seeds could be a pathway. *Dothistroma septosporum* has not been isolated from seeds, but inadequately cleaned seed material could contain infected needle fragments. This case study investigated whether cone kilning, and wet and dry heat treatments could reduce *D. septosporum* transmission without damaging seed viability. *Pinus* needles infected with *D. septosporum* were incubated alongside cones undergoing three commercial seed extraction processes. Additional needles were exposed to temperatures ranging from 10 to 67 °C dry heat for up to 48 h, or incubated in water heated to between 20 and 60 °C for up to one hour. *Pinus sylvestris* seeds were exposed to 60 and 65 dry heat °C for 48 h, and further seed samples incubated in water heated to between 20 and 60 °C for up to one hour. *Dothistroma septosporum* survived the three kilning processes and while seeds were not damaged by dry heat exceeding 63.5 °C, at this temperature no *D. septosporum* survived. Wet heat treatments resulted in less than 10% pathogen survival following incubation at 40 °C, while at this temperature the seeds suffered no significant impacts, even when submerged for one hour. Thus, commercial seed kilning could allow *D. septosporum* transmission, but elevated wet and dry heat treatments could be applied to seed stock to minimise pathogen risk without significantly damaging seed viability.

## 1. Introduction

The national and international trade in plants and reproductive material is commonly agreed to be a significant pathway responsible for the accidental movement of numerous invasive pests and pathogens between and within countries [1,2,3,4,5]. Legislation exists to minimise the threats posed by such organisms [6] but it struggles to keep pace with the volume and globalisation of traded goods [7,8]. Numbers of alien invertebrate species establishing annually in Europe are estimated to have doubled between 1950 and 2009 [9] whilst fungal invasives quadrupled between 1900 and 2009 [10]. In the USA, Liebhold et al. [11] estimate nearly 70% of the damaging forest pests which became established between 1860 and 2006 probably gained entry on imported live plants.

Seeds are more durable, smaller, and lighter than live plants, and are thus cheaper and easier to transport. They have received less attention as pathways for the movement of insect and fungal quarantine pests despite the high volumes of seeds traded [3,11,12]. Although 233 regulated organisms are currently associated with plants for planting in the EU, only one seed-borne pathogen of forestry significance, *Fusarium circinatum* Nirenberg & O’Donnell, is listed. Consequently, there are phytosanitary requirements for import and internal movement of *Pinus* spp. and *Pseudotsuga menziesii* (Mirb.) Franco seeds [5,13].

The world checklist of fungi includes over 550 seed-borne taxa [14]. Whilst many have no significant impact on seed or plant viability, some can damage seed quality and longevity [14,15]. A recent study of seeds traded between North America, Europe and Asia discovered fungal species present in all samples, 30% of them being potential gymnosperm pathogens [16]. In addition to *F. circinatum*, the introduction to, and movement within Europe of *Diplodia pinea* (Desm.) Kickx, Petrak & Sydow (syn. *Sphaeropsis sapinea* (Fr.) Dyko & Sutton), *Nematospora coryli* Peglion, and *Pestalotiopsis maculans* (Corda) Nag Raj. (syn, *P. guepinii* Desm.) can be at least partly attributed to dissemination on seeds [3].

Pathogens can be present on, and in the case of *F. circinatum*, cryptically within the seeds [13]. Viable propagules of pathogens including *Fusarium* spp. [17] and *Cladosporium* spp. [18] can also persist on debris accompanying inadequately cleaned seed material [19]. Certificates accompanying commercial seed material display the proportion of contaminants including inert matter which, although usually a small proportion of the seed stock, can contain fruit parts, wings, leaves, bark, stones, insect parts, nematodes, fungal bodies, soil, and seeds of other species [20].

Dothistroma needle blight (DNB) caused by *Dothistroma septosporum* (Doroguine) M. Morelet and *D. pini* Hulbary is a foliar disease with a worldwide distribution, predominantly on *Pinus* species [21]. Its spores are naturally disseminated via wind and rain [22], and the pathogen is also dispersed anthropogenically on infected plant stock [23,24,25]. Seed transmission is not considered to be a likely route as cones and seeds are often stored for several months, and the closed cones undergo heat treatments to extract seeds in a process called kilning [26]. Nonetheless, there is circumstantial evidence for the movement of *D. septosporum* on seed stock, as *D. septosporum* isolated from exotic *P. contorta* var. *latifolia* (Engelm.) Critchfield plantations in Scotland showed clear similarities with *D. septosporum* isolated from this host in its native range in British Columbia, Canada [27]. There are no records of large numbers of live plants being transported between the two countries, but over 9000 kg of seed was imported from western North America into Britain between 1920 and 1980, over 64% of this from British Columbia [28], with seed purity ranging from c. 88% to 99% in the 1930s and 1940s [29], (Figure 1). Previous studies have failed to find evidence of *D. septosporum* on seeds (27), but it is possible that infected foliar debris contaminating seed lots could present a potential pathway for the dissemination of *Dothistroma* species.

The temperatures used in cone kilning range between 30 and approximately 60 °C depending on factors including species and previous cone storage conditions. When developing a risk assessment for *Dothistroma* spp., the dry heat and associated decrease in seed moisture content were assumed sufficient to kill propagules of the pathogen on seeds or in any associated needle debris [26]. However, *D. septosporum* can persist for up to eight months on detached needles in the forest stand [30] and Ivory [31] reported *Dothistroma* spp. spores could remain viable in needle tissue incubated at 55 °C for ‘several days’, and survive desiccation at 30 °C for nine weeks. Seed lots contaminated with needle debris could, therefore, present a phytosanitary risk, as conditions experienced during kilning might not kill *Dothistroma* spp. in associated contaminant material.

In forestry, agriculture and the food industry, irradiation, microwave treatments, natural products such as essential oils, chemical and biological fungicides, inorganic chemicals including calcium hypochlorite, acetic acid, lactic acid, and fumigation with gases such as propylene oxide, chlorine dioxide, ozone, ethylene oxide, and ethylene chlorohydrin are used to control fungal and bacterial pathogens [32,33,34,35]. Due to concerns over the toxicity of some compounds [36,37] non-chemical hot-water immersion, steam sterilization, and hot air pasteurization can be effective when the pathogen is heat-sensitive and the seed relatively heat-tolerant [34,38,39,40,41]. As they penetrate the seed, they can target non-superficial infections [42,43]. Dry heat treatments can control *Fusarium* and *Alternaria* spp. [44,45,46]. Hot-water treatments, efficient as the thermal capacity of water is five times that of dry air [32,39], can eradicate bacterial pathogens *Salmonella* spp. and *E. coli*, *Fusarium* and *Alternaria* spp. [32,47], and the forestry pathogen *F. circinatum* [42,43]. It is therefore possible that additional dry air or hot-water heat treatments could also be effective against *Dothistroma* spp.

The effects of kilning and a range of dry and wet elevated temperatures and exposure times were examined in this study to determine what combination of temperature and time might kill *D. septosporum* within infected needles without significantly affecting seed germination. The specific objectives of this research were to (i) establish whether *D. septosporum* can persist on seeds, (ii) investigate whether kilning affects *D. septosporum* survival in needle tissue and (iii) determine temperature/time thresholds for dry or wet heat treatments required to kill *D. septosporum* in the needles without negatively impacting seed viability.

## 2. Materials and Methods

### 2.1. Persistence of the Pathogen in Seed Material

In 2022, seed-bearing cones were collected from stands of *P. sylvestris* L. planted in 1994, (51.1747, −0.8528) and *P. nigra* subsp. *laricio* (Poir.) Maire planted in 1967 (51.1768, −0.8588), both close to Alice Holt Research Station, England (51.1790, −0.852165). Prior inspections had demonstrated thin crowns and the characteristic visual symptoms of *D. septosporum*, red brown banding and small black fruit bodies on the foliage. Coning was sparse and only eight cones from three *P. sylvestris*, and five cones from two *P. nigra* subsp. *laricio* were sampled from trees with accessible crowns. Cones were left at room temperature for 1 week to enable seeds to be easily extracted as the cones opened. Each cone was split into cone scale material and seed material. Half of the cone scale material was crushed with a hammer sterilised between each cone sample by spraying with 10% bleach, then flaming with Industrial Methylated Spirits (IMS). The other half of the cone material was cut into sections small enough to fit into microcentrifuge tubes (MCTs), using a chisel sterilised between each cone sample with bleach and IMS as before. Half of the seed material from each cone was crushed, and half left intact. This resulted in four samples from each original cone—crushed cone scales, solid sections of cone scales, crushed seeds and intact seeds. Each of the four samples was further split into two; half of each sample was put into an MCT and the other half decanted into a 20 mL sterile plastic universal bottle containing 10 mL wash solution comprising one drop of Tween 80 (Sigma-Aldrich, St. Louis, MI, USA) in 50 mL sterile deionised water. The samples in the wash solution were shaken vigorously by hand for 30 s at intervals of 2–5 min for a total of ten 30 s shaking cycles following the methodology of Crosse [48] to remove inoculum. The liquid samples were filtered under vacuum, through MF Millipore 1.2 µm Gridded MCE 25 Filters which were then cut into smaller sections and put into MCTs. This method of sampling was carried out to distinguish between inoculum which might be present on the surface of seeds/cones versus inoculum which might be present within the body of seeds or cones.

The presence of *D. septosporum*, as well as *D. pini* and *Lecanosticta acicola* (Thümen) Sydow, the causal agent of brown spot needle blight, was tested using qPCR. DNA extraction was carried out using the MagMax Plant DNA Extraction Kit (Thermo Fisher Scientific, Basingstoke, UK) in a Kingfisher semi-automated extraction device (Kingfisher Flex, Thermo Fisher Scientific, UK) according to the manufacturer’s instructions. The quantitative real-time PCR reactions were carried out on a LightCycler 480 (Roche Diagnostics, Indianapolis, IN, USA), using the method developed by Ioos et al. [49], as carried out in Mullett et al. [22]. Each sample was run in triplicate with 18S and appropriate positive and negative pathogen DNA controls. 

### 2.2. Impacts of Kilning on D. septosporum Survival in Needle Material

Sections of needles with visible signs of *D. septosporum* infection were placed alongside cones undergoing commercial kilning to facilitate seed release, before extracting spores from the fruit bodies and testing the viability of the inoculum.

#### 2.2.1. Preliminary Tests of *D. septosporum* Viability in Needle Material

During the spring of 2015, five needles of each host species were collected from a mixed *P. sylvestris* and *P. nigra* subsp. *laricio* plantation established in 2005 near Farnham, Surrey, in Britain (51.184517, −0.850775). The trees were known to be affected by DNB and all needles had one or more mature fruit bodies. Spore presence and viability were confirmed by excising the fruit bodies, crushing on a glass slide with a droplet of sterile distilled water (SDW) and examining under ×400 magnification, before streaking onto duplicate 90 mm plates containing DM + S media (based on Dothistroma Medium [50] augmented with 2.5% of 1% streptomycin sulphate salt solution, Sigma-Aldrich). The plates were incubated at 20 °C for five to seven days before the *D. septosporum* colonies, exhibiting distinctive red/brown staining of the agar could be counted. Plates were examined again after another five days and presence or absence of *D. septosporum* colonies recorded on each plate. The molecular methods detailed above were used to confirm the presence of *D. septosporum* and additional needles were collected to examine the impacts of industrial cone-kilning treatments and controlled temperature–time treatments on *D. septosporum* viability (as stated below).

#### 2.2.2. Impacts of Forest Industry Cone-Kilning Processes on *D. septosporum* Viability

As the time taken for *Pinus* cones to open and release seeds depends on the species, provenance, and cone storage conditions, kilning conditions can vary slightly. Infected *P. sylvestris* and *P. nigra* subsp. *laricio* needles were exposed to three kilning treatments termed ‘SE1’ and ‘SE2’, and ‘SE3’, replicating conditions used by commercial British seed production companies (see Table 1).

Process SE1 consisted of keeping needle material from the two host species separate; the pine needles were sealed into paper envelopes, and placed inside wooden crates containing cones undergoing the 74 h long kilning process SE1, which took place in an industrial tunnel oven. TinyTag Probes (TinyTag View 2 TV-4020; Gemini Data Loggers, Chichester, UK) monitored temperature and humidity every five minutes, recording a maximum of 51.6 °C during the day, and a minimum 20.0 °C overnight when the ovens were not operating due to noise restriction measures. After 74 h, the cones were moved into a heated rotating drum for one hour. The envelopes containing the needles and the TinyTag probe, which recorded a maximum temperature of 38.7 °C, were firmly attached by wires to the side of the drum. Seeds released from the cones by the rotation and agitation fell through small holes in the drum and were collected in a tray underneath before being stored in an air-tight container at 5 °C. The needles were stored at 10 °C overnight before further testing as outlined below. 

In process SE2, cones and bagged infected needles (as above) were incubated in the tunnel oven, together with a TinyTag probe for 30 h, reaching a maximum of 49.8 °C, and minimum of 24.6 °C overnight. After this time, seeds and cones were incubated in the heated rotating drum for one hour at a maximum temperature of 36.9 °C, after which seeds and needles were collected and stored as above before further testing.

In process SE3, infected needles were sealed, separately by host species, inside paper envelopes and incubated at 40 °C in an incubator (Sanyo MIR-253 Cooled Incubator, Sanyo Electric Co., Moriguchi, Osaka, Japan) for 48 h, followed by 17 h at 35 °C before further testing (see Table 1).

Following the three kilning processes, a single fruit body from each needle was excised under a dissecting microscope and crushed on a glass slide with a droplet of SDW. The presence of spores was confirmed visually as described above, and samples of spores from each sample streaked onto duplicate DM + S plates before incubating at 20 °C and the presence/absence of germinating spores was quantified on each plate after 10–12 days. 

### 2.3. Effect of Controlled Dry Heat Treatments on D. septosporum and Seed Viability

Infected needles and *P. sylvestris* seeds from a commercial British seed supply company were exposed to a range of controlled dry heat treatments using incubators (Sanyo MIR-253 Cooled Incubator, Sanyo Electric Co., Japan) programmed to run at temperatures ranging from 20 to 67 °C as follows.

#### 2.3.1. *Dothistroma septosporum* Viability in Needles (Dry Heat)

Samples of five infected needles from each *Pinus* host species were sealed inside paper bags and laid in metal trays to incubate at 10, 20, 40, 50, 55, 60, 63.5, or 67 °C for 48 h.

After this time, a single fruit body was excised from each of the five needles per host/temperature treatment. The presence of spores and spore viability, and growth after 10–12 days was assessed as described above.

#### 2.3.2. Seed Viability (Dry Heat)

*Pinus sylvestris* seeds extracted using process SE1 were obtained from a commercial British cone-processing company. All seeds were stored at −5 °C before use, and one control sample of 30 g of seeds maintained at this temperature while the remaining seeds were exposed to the following time–temperature treatments. Aliquots of eight grammes of seeds were placed in tins in incubators set at 60 and 65 °C, and incubated for three time periods (6, 24 or 48 h), resulting in six dry heat treatment combinations. The moisture content (MC), and viability of seeds before and after treatment were assessed using standard methods as follows. 

Moisture content—The % MC of seeds in each sample was determined on a fresh weight basis using the low-constant-temperature oven method (17 h at 103 °C [51]). 

Germination tests—Two sets of one hundred seeds from each treatment combination were placed on moistened filter paper (Sartorius Stedim Keimtestpapier 300 gm^−2^) in germination boxes as described by Gosling [51]. They were subjected to double-germination tests, that is, with and without a 3-week chill between 4 and 5 °C. One set of 100 seeds was placed in a moist environment at 3 to 5 °C for 21 days to pre-chill in a germination box before entering germination tests [20]. The other set of 100 seeds from each treatment combination was immediately subjected to germination tests and then incubated at 20 °C (constant dark) for up to four weeks post treatment, and germination assessed every three days [52]. Seeds were considered germinated when the radicles were four times longer than the seed. Seeds were categorised as germinated, live (but ungerminated), empty or dead/abnormal. Abnormal germinants included breached seedlings and seedlings with stunted growth and brown radicles [51] (Figure 2). The un-germinated seeds were cut open and examined visually before being able to assign them to the live, dead, or empty categories. Empty seeds, where the seed contents occupy less than 50% of the cavity, can result from the abnormal development of the female gametophyte or when an embryo aborts. ‘Live’ seeds are viable, showing some expansion of embryonic tissue, but fail to rupture the seed coat and germinate within the experimental timeframe, whilst ‘dead’ seeds fail to develop and were either dead at the outset or died because of the imposed treatments.

### 2.4. Effect of Wet Heat Treatments on D. septosporum and Seed Viability 

#### 2.4.1. *Dothistroma septosporum* Viability in Needles (Wet Heat)

Incubators were programmed to run at 20, 40, 50, 55 and 60 °C, the temperature recorded every five minutes using TinyTag data recorders (TinyTag View 2, Gemini Data Logger Ltd., Chichester, UK) with one probe resting on the incubator shelf and a second probe fully submerged in a glass vial containing 50 mL SDW. 

For each temperature treatment, 30 glass vials were filled with 50 mL SDW and allowed to equilibrate for one hour. Fifteen infected needles of each of the two host species were incubated at each temperature by fully submerging a single infected needle in each vial. Five needles of each host species were incubated at each of three exposure times, 15, 30 and 60 min, resulting in 30 treatments comprising the two host species, five temperatures and three exposure times.

After each temperature/time treatment, the needles were removed from their individual vials of water and laid separately on damp paper towels. A fruit body was excised from each needle and the presence and viability of spores confirmed as detailed previously. In addition, each vial of SDW was shaken for 20 s and two additional DM + S plates were individually inoculated with 1 mL of the liquid in which the needle had been suspended, to assess the viability of any spores exuded from fruit bodies into the surrounding liquid. All plates were incubated at 20 °C for 10–12 days and presence/absence of colonies was recorded as before.

#### 2.4.2. Seed Viability (Wet Heat)

A 30 g sample of *P. sylvestris* seeds was retained as the control treatment, with exposure time of 0 min and a temperature of 20 °C. Water baths (each 32 L), fitted with thermostatically controlled immersion heaters, were set to four different temperatures (20, 40, 50 and 60 °C), the temperatures recorded using TinyTag recorders as before. Four 1 L glass flasks filled with SDW were left to equilibrate in each water bath for an hour, after which three beakers, each containing 30 g of *P. sylvestris* seeds, were added to three of the flasks, one beaker for each of three exposure times (15, 30 and 60 min). An additional temperature-recording probe was placed inside the fourth flask. 

After each temperature–time treatment, the seeds were plunged in cold tap water for two minutes before being dried on paper towels to remove surface water. Four samples, each containing 100 seeds, from each treatment combination, were subjected to MC, and germination analysis as described above, to determine seed thermotolerance.

### 2.5. Statistical Analysis

All statistical tests were performed using R (version 4.2.1) [53] and were considered significant at *p* < 0.05.

#### 2.5.1. Analysis of *D. septosporum* Survival in Infected Needles

Data from needles exposed to kilning treatments and the dry heat treatments were combined to form a ‘kilning/dry heat’ dataset with a temperature range of 10 to 67 °C. Petri dishes with numbers of colonies exceeding zero were assigned a value of 1 while plates without colony growth were assigned a value of zero. The data were subsequently aggregated, such that scores of 0, 1 and 2 were possible (equivalent no growth on either plate, growth on one plate only, or growth on both plates). The wet heat treatments formed a separate dataset, but data were assigned scores in the same way.

Generalised linear mixed effects models were fitted to the data, with binomial errors and logit link functions, with the response variable being the presence/absence of growth of colonies. Model fit was assessed using model simulation data (DHARMa package) [54]. For the kilning/dry heat dataset, host species, treatment/temperature and their interaction were included as factors within the model. The significance of factors was determined based on the likelihood-ratio chi-square test statistics from the analysis of deviance, using the car package in R [55]. Where possible, post hoc tests were used on the best fit model to estimate differences within significant factors, correcting for multiple comparisons using Bonferroni adjustments to the *p* value. In some cases, responses were all 0 s or 1 s for a particular treatment; in these cases, accurate contrasts were not possible, but confidence intervals were estimated using exact binomial tests, with 95% confidence intervals adjusted for multiple comparisons using Bonferroni corrections.

For the wet heat dataset, species, whether potential colonies originated from the fruit body or the surrounding water, temperature, and time, plus all interactions were included as factors within the model. The significance of factors was determined based on the likelihood-ratio chi-square test statistics from the analysis of deviance, using the car package in R [55]. Where possible, post hoc tests were used on the best fit model to estimate differences within significant factors, correcting for multiple comparisons as above. In some cases, responses were all 0 s or 1 s for a combination; in these cases, accurate contrasts were not possible, but confidence intervals were estimated using exact binomial tests.

#### 2.5.2. Analysis of Seed Viability following Treatments

For the dry heat treatments, seeds exposed to each temperature and exposure time treatment were classed as ‘viable’ (which included seeds in the early stages of germination), or ‘empty’ or ‘dead’. All empty seeds were removed from the dataset prior to analysis as their capacity to germinate was unaffected by the heat treatments. The influence of main-effect temperature, exposure time and pre-chill, and their interactions, on the proportion of viable seeds, were assessed using analysis of variance (ANOVA) with appropriate residual analysis to check for normality [55]. Further comparisons of treatment effects were carried out using a Tukey HSD test.

In the wet heat dataset, seeds classified as empty were removed from the dataset prior to analysis as before. Seeds in the remaining dataset set were classed as viable (i.e., germinated + ‘live’), abnormal, or dead. Differences in the proportion of viable seeds over those which were dead/abnormal were analysed using a generalised linear model with a Poisson distribution, and ANOVA as above, using Tukey’s HSD test to perform further between-treatment investigations.

## 3. Results

### 3.1. Persistence of D. septosporum in Seeds

Analysis of all solid and crushed seed and cone scale samples, and all surface washing samples, tested negative for *Dothistroma* spp. and *L. acicola* using the multiplex qPCR bioassay.

### 3.2. Impacts of Kilning and Dry Heat Treatments on D. septosporum Survival and Seed Viability

#### 3.2.1. *Dothistroma septosporum* Viability in Needles—Kilning and Dry Heat

One hundred percent of samples of untreated, symptomatic *P. sylvestris* and *P. nigra subsp. laricio* needles yielded viable colonies of *D. septosporum* when spore suspensions from fruit bodies were inoculated onto DM + S. The growing cultures were confirmed to be *D. septosporum* using the molecular techniques described in Section 2.1.

An analysis of kilning and dry heat treatments on *D. septosporum* survival *in planta* demonstrated that there was a significant effect of temperature on the presence/absence of viable *D. septosporum* on plates inoculated with spores from the colonised needles (Χ^2^ = 151.43, df = 8, *p* < 0.001; Figure 3), with decreasing survival at increasing temperatures. There was no significant effect of host species (Χ^2^ = 0.07, df = 1, *p* = 0.789) or the interaction of host species and treatment/temperature (Χ^2^ = 9.73, df = 8, *p* = 0.285). No *D. septosporum* colony growth was observed in samples taken from needles incubated at temperatures of 63.5 °C or above, whilst viable colonies were present in samples exposed to all lower temperatures (Figure 3). Viable spores persisted within needles exposed to all three cone-kilning processes. 

#### 3.2.2. Seed Viability—Kilning and Dry Heat

All seeds used in the dry heat bioassay were processed using the SE1 process, and had an initial viability of 68% (non-pre-chilled) and 73% (following a 3-week pre-chilling period).

Moisture content—Before treatment, MC was 7.9%. After incubation at elevated temperatures, the seeds lost more water, with MC generally decreasing with increasing temperature and longer exposure time, reaching a low of 2.0% after 48 h incubation at 65 °C (Figure 4). 

Germination—Although pre-chilling increased the germination of untreated seeds, after exposure to the higher temperature, the pre-chill treatment appeared mostly to reduce seed viability (F_(1,42)_ = 9.72, *p* < 0.01). Viability was also influenced by temperature and exposure time (F_(2,42)_ = 6.19, *p* < 0.05). Two days’ incubation at 60 °C had little influence on seed behaviour but the most pronounced treatment effect was apparent after seeds were incubated for 48 h at 65 °C then subjected to the pre-chill treatment (*p* < 0.001), reducing viability from 73% pre-treatment to 49% (Figure 5).

### 3.3. Impacts of Wet Heat Exposure on D. septosporum Survival in Needles and Seed Viability

#### 3.3.1. *Dothistroma septosporum* Survival in Needles—Wet Heat

The host species had a significant influence on the presence of viable spores following wet heat exposure (Χ^2^ = 8.63, df = 1, *p* = 0.003), with a greater likelihood of finding viable spores in *P. nigra* subsp. *laricio* compared with *P. sylvestris* needles. The temperature treatment also significantly affected viable spore presence (Χ^2^ = 234.83, df = 4, *p* < 0.001), with viable spore presence decreasing with increasing temperature (Figure 6). Although exposure time was not a significant main effect (Χ^2^ = 1.96, df = 2, *p* = 0.376), the interaction of temperature and exposure time was significant (Χ^2^ = 48.13, df = 8, *p* < 0.001; Figure 6): at the shortest time interval, both 20 °C and 40 °C had a significantly higher presence of colonies than other temperatures. However, at longer time intervals, only 20 °C showed significantly higher colony presence (*p* < 0.001). Very low survival (<10%) was observed at temperatures higher than 40 °C and no *D. septosporum* spores germinated at the highest incubation temperature (60 °C).

#### 3.3.2. Seed Survival—Wet Heat

Moisture Content—The initial MC of 7%, increased after the wet heat treatment, as the seeds imbibed water. MC increased with increasing temperature and exposure time from a minimum of 17% after 15 min incubation at 40 °C, to a maximum of 32% after 60 min at 60 °C (Figure 7).

Germination—Seed viability was significantly affected by the temperature, incubation time and pre-chill treatments (Figure 8). Exposure to increasing temperatures decreased seed viability, but the effect was more pronounced in seeds then exposed to a pre-chill (F_(3,78)_ = 6.13, *p* < 0.001). A significant incubation time x temperature interaction was also apparent (F_(6,78)_ = 8.91, *p* < 0.001): exposure to 50 °C for 15 min had no significant effect (*p* = 1.00), whilst longer incubation of 60 min at 50 °C caused a 27% and 56% decline in viability in non-pre-chilled and pre-chilled seeds, respectively, and 60 min exposure to 60 °C killed all non-pre-chilled seeds and reduced the viability of the pre-chilled seeds to less than 10% of the untreated controls (Figure 8). Further examination of the seeds revealed an increase in seed mortality with increasing temperatures, but also a rise in number of damaged seeds which produced abnormal seedlings. 

## 4. Discussion

This study provides further confirmation that *D. septosporum* does not appear to be transmitted via seeds [26], as no traces of the pathogen were found in or on the cone and seed material collected from an English forest affected by Dothistroma needle blight. Thus, pure seed should not be considered a pathway for the transmission of this pathogen. 

However, an examination of archived seed records going back to 1934 demonstrated that *Pinus* seed purity ranged from approximately 88 to 100% [29] (Figure 1). Even the seed stock used in this trial contained 7.4% inert material, mostly in the form of small stones and pieces of cone material. As neither historical nor current methods of seed preparation appear to remove all associated contaminants, including, potentially, infected sections of pine needles, it was important to establish whether *D. septosporum* could survive kilning processes used by industry, and whether additional treatments could be effective, should viable spores be present in the debris associated with the seed material.

The inclusion of infected pine needles with cones undergoing commercial kilning processes demonstrated that *D. septosporum* could survive temperatures of up 52 °C in planta over a 72 h period, with no significant decrease in viability. The European risk assessment considered 30 °C the lethal limit for *D. septosporum* growth in vitro [56], and c. 35 °C for the similar foliar pathogen, *L. acicola* [57]. If these temperature thresholds were correct, transmission via the ‘fourth route’, i.e., on seed stock, would be unlikely [26]. However, Ivory successfully re-isolated *Dothistroma* spp. after incubation at far higher temperatures [31]. There was an additional assumption in the risk assessments that storing cones before kilning to allow further maturation or lower moisture content to reduce subsequent kilning costs [58] might reduce the viability of *D. septosporum* in the associated debris. However, Ivory successfully isolated *Dothistroma* spp. from herbarium samples kept at room temperature for 11 months [31], and Mullett et al. [30] retrieved the pathogen from fallen needles after eight months. Thus, several months of cone storage might not substantially affect the viability of *D. septosporum* in any needle debris.

As standard cone-processing and -kilning processes cannot be guaranteed to kill *D. septosporum* in infected needle debris, this study investigated further heat treatments. *Dothistroma septosporum* was shown to have a lethal limit *in planta circa*. 63.5 °C in both *P. nigra* subsp. *laricio* and *P. sylvestris* needles. The *P. sylvestris* seeds retained pre-treatment levels of viability even after 48 h incubation at 60 °C, and after short periods held at 65 °C, although viability declined with longer incubation at this highest temperature. For logistical reasons the seeds were exposed to a slightly reduced range of temperatures than those to which the *D. septosporum*-infected needles were exposed. In any future work, elevated temperatures would be examined at a higher resolution, to determine precise temperatures sufficient to kill *D. septosporum*, without damaging seed viability significantly. It would also be vital to examine whether any of the heat treatments had longer-term impacts on stored seed viability. Any differences in the tolerance of varying sized seeds was not investigated, and it would also be important to test seed from a range of species, in particular *P. contorta* var. *nigra*, on which it is hypothesised that *D. septosporum* may have travelled between British Columbia and Britain. However, seeds of this species were in short supply. Overall, however, the higher apparent tolerance of the *P. sylvestris* seeds than the pathogen *D. septosporum* to elevated temperatures demonstrates the potential for the use of dry heat to combat the pathogen in dry seed.

Wet heat treatments had a more immediate impact on *D. septosporum* survival, with less than 10% pathogen survival following incubation at a relatively low temperature (40 °C water for more than 15 min). The reasons for the slightly higher survival of *D. septosporum* in *P. nigra* subsp. *laricio* compared with *P. sylvestris* is not clear. At 40 °C, wet-heat-treated *P. sylvestris* seeds suffered no significant impacts on viability, even when submerged for up to one hour, with the result that this wet heat treatment could potentially be used to deactivate pathogen inoculum in associated needle debris. 

Higher temperature wet heat treatments tended to cause more damage to the seeds than the equivalent dry heat temperature. This is likely to be due to differences in seed MC, which was far higher after the wet heat treatments. Cones with a higher initial MC are ideally processed at lower temperatures, or treated to a more gradual rise in kilning temperature, to avoid damaging viability [58]. Physiological processes in seeds are strongly influenced by MC [59], and while the biological and physical unavailability of water in seeds with very low MC impedes respiration, enzymatic processes within the seeds can re-commence, and respiration rates increase when MC increases [60]. The available water within seeds incubated at the higher wet heat treatments was probably heated to a point where it damaged the cellular processes, denaturing enzymes, leading to increased mortality. 

Germination of gymno- and angiosperm seeds is often increased by a short pre-chill [61]. Although *Pinus* species seeds can exhibit shallow dormancy, and a pre-chill is often used [20], treatment response is influenced by genetic and environmental factors [38,62,63]. The *P. sylvestris* seeds used in this experiment possibly originated from different seed lots, as the control, i.e., baseline, untreated seeds used for the dry heat treatment bioassay benefitted slightly from a pre-chill, whilst pre-chilling seeds used as a control in the wet heat bioassay had a deleterious impact. A pre-chill appeared to lower the viability of seeds exposed to both wet and dry elevated temperatures, which may be due to a loss in seed vigour during the heat treatments.

Despite some uncertainties, these results show the promise wet and dry heat treatments could have in eliminating *D. septosporum* from needle debris. Whereas the temperature thresholds to kill other significant pathogens including *E. coli*, *Fusarium, Alternaria* and *Diaporthe* spp. have sometimes exceeded the heat tolerance of the seed material [32,64], *P. sylvestris* tolerated conditions that killed the majority of *D. septosporum* spores. This is not, perhaps, surprising as this species, although not as heat-tolerant as certain southern Mediterranean *Pinus* species originating from wild-fire-prone regions, can survive brief exposure to temperatures up to 130 °C [65]. Of the other *Pinus* species commonly planted in Britain, *P. contorta* var. *latifolia* is also known to be able to tolerate 69 °C for 8 h during cone processing [66], and it is likely that *P. nigra* subsp. *laricio* would show a similar level of tolerance, although this would need to be tested. 

The efficacy of heat treatments against other foliar pathogens should now also be investigated. Neither *L. acicola* nor *D. pini* are thought to be present in Britain, but both have the potential to cause significant damage to British and European forests [21,67,68]. Currently, far less is known about their capacity to ‘hitch-hike’ on needle debris, but the distribution of both pathogens is known to have expanded rapidly in the past half a century and is a cause for concern [69]. 

It appears that seeds, although presenting less risk than plants for planting, or timber products, are not necessarily as safe as previously thought. There is increasing evidence for the global movement of many pathogens on seed stock. In the case of *D. septosporum*, the evidence for movement on seeds from British Columbia to Britain is circumstantial. *Dothistroma septosporum* is not transmitted vertically on seed stock, in the same way as, for example, *F. circinatum*. However, the pathogen clearly can survive for considerable periods in detached needles (Ivory [31] and Mullett et al. [27]). Additionally, this study demonstrates it can also survive the temperatures imposed by commercial seed extraction processes. The actual mechanism by which *D. septosporum* spores or mycelium from infected needle material could have transitioned from poorly cleaned seed stock onto growing seedlings in a nursery or plantation remains unclear, and yet at a molecular level, populations of *D. septosporum* in Britain bear a striking resemblance to those from where significant quantities of seed stock were imported [27].

Given the difficulties in detecting fungal pathogens in or on seed stock during border inspections, the safest approach is to combine better seed trade regulation with more efficient seed-processing techniques and effective treatments to eliminate or reduce pathogen viability in seed stock [16]. The wet and dry heat treatments explored in this study showed considerable promise in being able to close down this pathway in the future.

## 5. Conclusions

The results from this study support the view that *D. septosporum* is not transmitted in or on seeds. Thus, clean seeds do not form a pathway for the transmission of this pathogen. However, standard cone-kilning treatments do not expose any needle debris accompanying poorly cleaned seeds to temperatures hot enough to kill *D. septosporum in planta*. Additional dry heat treatments over 63.5 °C and or wet heat treatments of 40 °C between 30 to 60 min or 50 °C for 15 min can be used to kill *D. septosporum* on poorly cleaned *P. sylvestris* seed stock containing needle fragments. *Pinus sylvestris* seeds could withstand temperatures of 60 °C (dry heat) and 40 °C for (wet heat) without a loss in viability. Further research would be needed to investigate any longer-term impacts of heat treatments on stored seed viability, to determine the tolerance of seed stock from other *Pinus* species to the heat treatments and to establish the thermotolerance of closely related pathogens including *D. pini* and *L. acicola*, both of which present a significant potential risk to British forests.

## Figures and Tables

**Figure 1 jof-09-01190-f001:**
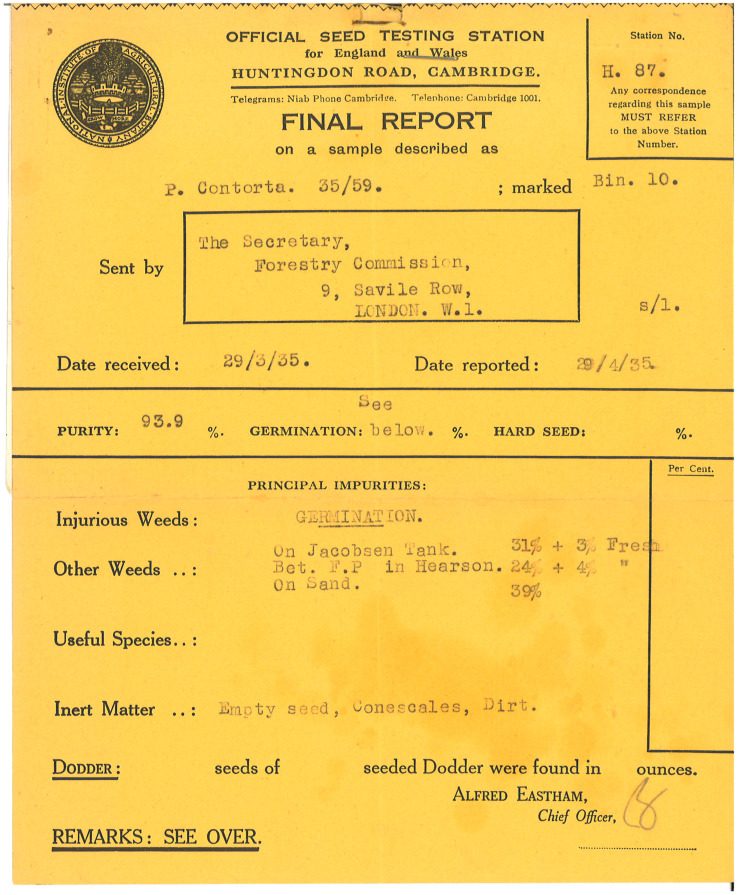
Example of historic seed purity certificate for *P. contorta* seeds tested in 1935, showing a purity of 93.9% and a description of various contaminants.

**Figure 2 jof-09-01190-f002:**
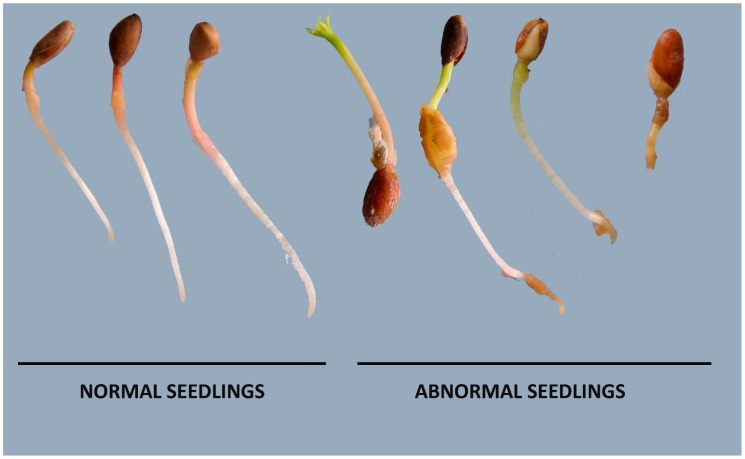
Photographs illustrating germination of normal and abnormal seedlings. In all subsequent analyses, abnormal seedlings are included with dead seeds when investigating impacts of dry and wet heat treatments on seed viability.

**Figure 3 jof-09-01190-f003:**
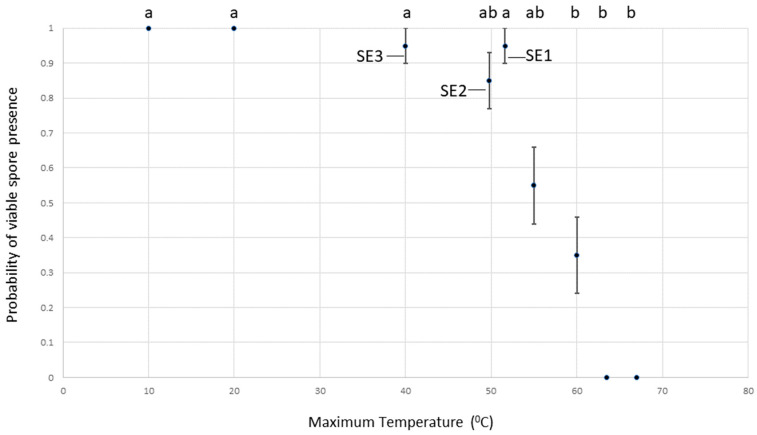
Effect of kilning (SE1, SE2 and SE3 protocols) and elevated dry heat treatments on probability of *D. septosporum* spore viability. Treatments with same lower-case letters did not differ significantly (Tukey, HSD, *p* > 0.05). (Error bars indicate standard errors).

**Figure 4 jof-09-01190-f004:**
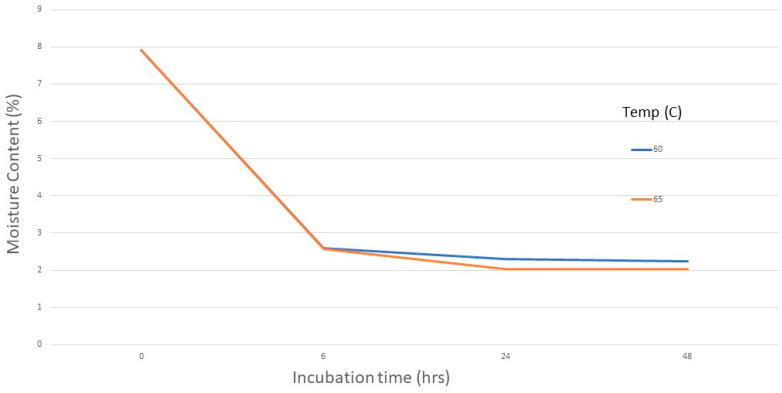
Moisture content of seeds after incubation at 60 or 65 °C (dry heat) for up to 48 h.

**Figure 5 jof-09-01190-f005:**
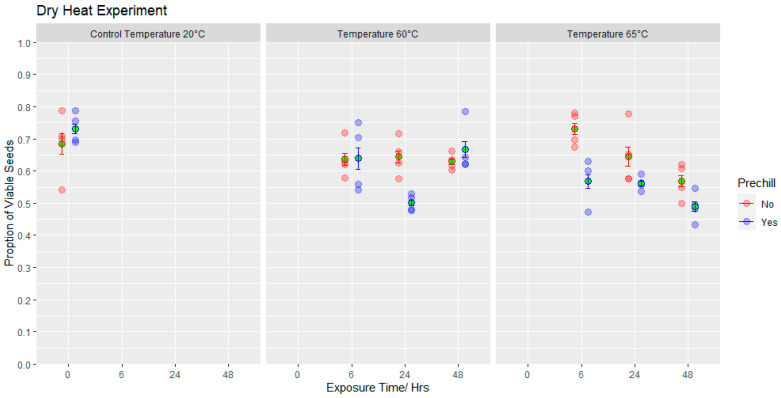
The proportion of seeds which were viable following exposure to dry heat treatments compared with the viability of untreated seeds after a 3-week pre-chill or no pre-chill (mean values shown in green, error bars display 95% confidence intervals).

**Figure 6 jof-09-01190-f006:**
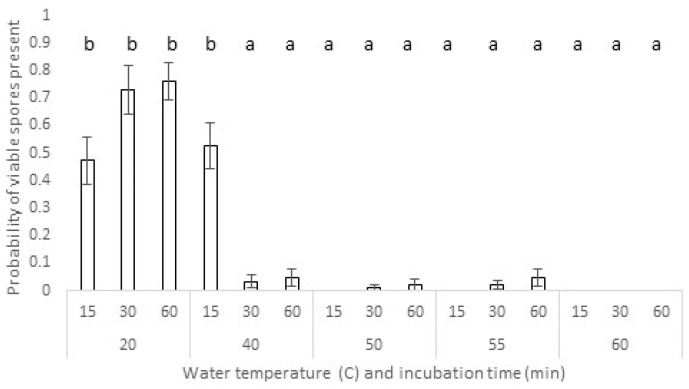
Impact of hot-water treatments on survival of *D. septosporum* spores after incubation in water at a range of temperatures over 3 time periods. Treatments with same lower-case letters did not differ significantly (Tukey, HSD, *p* > 0.05). (Error bars indicate standard errors).

**Figure 7 jof-09-01190-f007:**
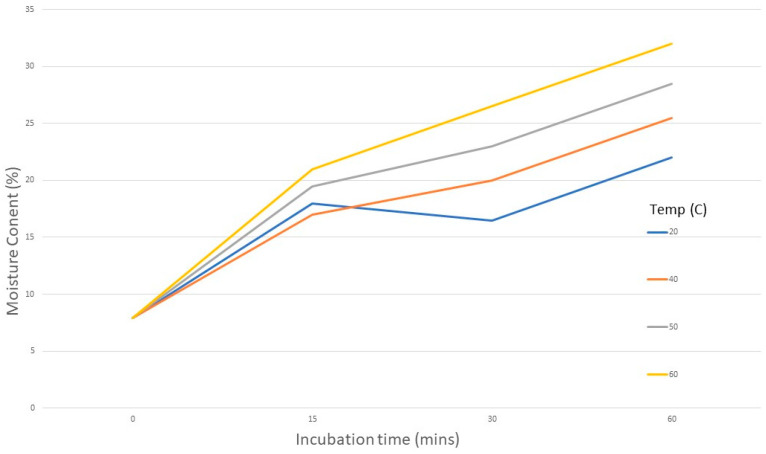
Moisture content of seeds (%) before and after incubation in water at 4 temperatures for up to 60 min.

**Figure 8 jof-09-01190-f008:**
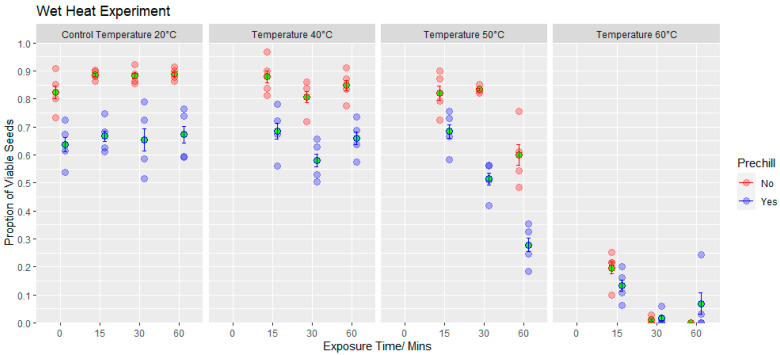
A comparison of seed viability following exposure to a range of hot-water exposure treatments ranging from 20 to 60 °C, for time intervals from 0 to 60 min and after exposure to a 3-week pre-chill or no pre-chill (means shown in green, error bars display 95% confidence intervals).

**Table 1 jof-09-01190-t001:** Temperature and times experienced during three different kilning processes. SE1 and SE2 were the temperature–times recorded during two cone-kilning processes at one commercial seed-processing company, while SE3 was a reconstruction of a process used by a second seed company, simulated in a temperature-controlled incubator (Sanyo MIR-253 Cooled Incubator, Sanyo Electric Co., Japan).

Kilning Process		SE1	SE2	SE3
Heating: step 1	Time (h)	74	30	48
	Temperature (°C) during heating cycle	Mean: 49.9 Min: 46.6 Max: 51.6 s.e. = 0.09 (25.0 overnight min *)	Mean: 48.5 Min: 47.0 Max: 49.8 s.e. = 0.07 (24.6 overnight min *)	Mean: 39.0 Min: 37.5 Max: 40.2 s.e. = 0.014
Heating: step 2	Time (h)	N/A	N/A	17
	Temperature min/max (°C)	N/A	N/A	34.3–34.8 Mean = 34.6 s.e. = 0.007
Seed cleaning	Time (h)	1 **	1 **	N/A ***
	Mean temperature (°C) (min/max)	Mean: 37.6 Min: 35.0 Max: 38.7 s.e. = 0.34	Mean: 35.3 Min: 32.1 Max: 36.9 s.e. = 0.43	N/A ***

* kiln ovens switched off overnight, temperature drops to ambient levels. ** debris removed by agitating cones/seeds in heated rotating drum. *** debris removed manually by pushing seeds through coarse sieve. Pine needles undergoing process SE3 not exposed to this treatment.

## Data Availability

The datasets generated and analysed during the current study are deposited in the FR repository, and are available upon reasonable request.

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
