# Peer review of "Can the Seed Trade Provide a Potential Pathway for the Global Distribution of Foliar Pathogens? An Investigation into the Use of Heat Treatments to Reduce Risk of Dothistroma septosporum Transmission via Seed Stock"

_jof, 2023, doi:10.3390/jof9121190_

Round 1

Reviewer 1 Report

Comments and Suggestions for Authors

This article aimed to investigate whether cone kilning and wet and dry heat treatments could reduce D. septosporum transmission without damaging seed viability. The authors indicated that commercial seed kilning could allow D. septosporum transmission, but elevated wet and dry heat treatments could be applied to seed stock to minimise pathogen risk without significantly damaging the seed viability.

The article is straightforward, and it contains original information. Throughout the article, the reviewer suggests the authors use the abbreviation of the pathogen term (i.e., D. septosporum) except when it appears for the first time. Shortening the background information and including an elaborated but brief methodology description (i.e., treatment temperature and time) in the Abstract section would improve the article. The authors may also consider changing the title to align with the specific research conducted. 

In addition, this article would be improved if the authors revised and clarified the following: 

Line 83. Revise to “[27].”

Lines 164, 225, 272, 352, and 397. Non-italicize to “D. septosporum.”

Lines 168, 184, 231, 273, 308357, 398, and 441. Italicize to “D. septosporum.” 

Line 169. Wouldn’t they (2015 samples) be too old for this type of study?

Line 212. Revise to “Temperatures and times ….”

Table 1. Revise to “°C.”

Line 227. Remove “also.”

Line 237. May revise to “described below.”

Line 263. Revise to “50%.”

Line 265. How did the authors determine “viable”?

Line 296. Remove “C.”

Line 310. Revise to “°C.”

Line 361. Where is “the molecular techniques described above”? The authors need to be specific about the location where the techniques are described.

Figure 3. Statistical significance annotations do not match with the error bars.

Line 373. Remove “_.”

Lines 386-387. Revise to “Although there was increasing germination … the higher temperature, the pre-chill ….”

 Line 410. Revise to “temperature (60°C).”

Lines 453-456. Revise the sentence for clarification.

Lines 460-462. Revise to “Ivory [31] successfully isolated ….”

Line 466. Revise to “… were investigated in this study.”

Lines 470-472. What do the authors try to say in this sentence? Revise the sentence for clarification.

Lines 473-474. Revise the sentence to “to determine precise temperatures sufficient to kill ….”

Lines 478-479. May revise the sentence to “…during this study period.” for clarification.

Lines 483-484. Revise to “… low temperature (40°C) water ….”

Line 532. Revise to “needles (Ivory [31] and Mullett et al. [27]). Additionally, this study demonstrates that it can ….” 

Line 539. Revise to “in detecting fungal pathogens in the form of spores or mycelia in or on seed stock during ….”

Lines 547-548. Revise to “However, standard cone … in planta (please confirm if this is the right term).”

Comments on the Quality of English Language

Minor editing of English language required.

Reviewer 2 Report

Comments and Suggestions for Authors

Authors studied whether cone kilning, and wet and dry heat treatments could reduce Dothistroma septosporum transmission without damaging seed viability.

There are too many references, statement about international trade as pathway of pathogens does not require any reference. (Lines 35 – 37)

Fungal names should be written without authors, authors are necessary in studies related to taxonomy.

I do not think, that data about Fusarium sp., Cladosporium significantly improve explanation of this research soundness. (Lines 57 – 67)

Figure 1 and Figure 2 are not necessary.

The manuscript is clear. Methods of research are described in details, but arrangement is too fragmented, no necessary so many chapters and sub-chapters, similarly about “Results”.

Conclusions clearly summarize obtained data.

Reviewer 3 Report

Comments and Suggestions for Authors

Dothistroma needle blight, also known as red band needle blight, is a very serious needle disease of conifers caused by fungi Dothistroma septosporum.

 The authors did a lot of research work. However, the manuscript requires a serious review. In this form, its publication can cause damage in the practice of seed harvesting. An increase in temperature above used in practice is dangerous for further preservation of the germination of seeds for a long storage period. The germination in the soil is also much more difficult than in laboratory conditions. It will also be difficult to support the temperature and humidity of the seeds recommended by the authors. Significant variation of seeds in size within the same type also creates problems with the choice of maximum temperature.

The authors rightly indicate «This study provides further confirmation that D. septosporum does not appear to be transmitted via seeds».  Therefore, in practice, everything can be solved differently. Firstly. Strengthen the requirements for cleaning seeds from garbage. Secondly. Consider additional treatment with fungicides. Third. Given «D. septosporum can persist for up to eight months on detached needles in the forest stand [30]»  to enter seeds to move seeds.

It would be interesting to investigate the fungicidal properties of the soil in relation to D. septosporum.

Reviewer 4 Report

Comments and Suggestions for Authors

Please find my comments in the attached pdf file.

Good luck

Round 2

Reviewer 3 Report

Comments and Suggestions for Authors

Taking into account the corrections and answers to comments, I consider it advisable to accept in the present form